The effects of different rootstocks on aroma components, activities and genes expression of aroma-related enzymes in oriental melon fruit

Guo Kedong 1 2
Zhao Jiateng 1 2
Fang Siyu 1 2
Zhang Qian 1 2
Nie Lanchun yynlc@hebau.edu.cn 1 2 3
Zhao Wensheng zhaowensheng@hebau.edu.cn 1 2 3
1 College of Horticulture, Hebei Agricultural University , Baoding , Hebei , China
2 Hebei Key Laboratory of Vegetable Germplasm Innovation and Utilization , BaoDing , Hebei , China
3 Collaborative Innovation Center of Vegetative Industry of Hebei Province , BaoDing , Hebei , China
Kumar Ravinder
Electronic publication date: 2024 Jan 5
Publication date: 2024
Volume: 12
Electronic Location ID: e16704
Received 2023 Aug 30; Accepted 2023 Nov 30
Copyright: ©2024 Guo et al.
Copyright year: 2024
Copyright holder: Guo et al.
License: This is an open access article distributed under the terms of the Creative Commons Attribution License, which permits unrestricted use, distribution, reproduction and adaptation in any medium and for any purpose provided that it is properly attributed. For attribution, the original author(s), title, publication source (PeerJ) and either DOI or URL of the article must be cited.
License URL: https://creativecommons.org/licenses/by/4.0/

Keywords: Cucumis melo L., Rootstock, Aroma components, HS-SPME-GC-MS, Aroma-related enzymes, Gene expression

Funding: Science and Technology Project of Hebei Education Department ZD2020343 Hebei Protected Vegetables Innovation Team HBCT2023100208 National Natural Science Foundation of China 32002063 Natural Science Foundation of Hebei Province C2021204033 Special Research Foundation of Hebei Agricultural University YJ201837 This work was supported by Science and Technology Project of Hebei Education Department (ZD2020343), the earmarked fund for the Hebei Protected Vegetables Innovation Team from the Modern Agro-industry Technology Research System (HBCT2023100208), the National Natural Science Foundation of China (32002063), the Natural Science Foundation of Hebei Province (C2021204033), and the Special Research Foundation of Hebei Agricultural University (YJ201837). The funders had no role in study design, data collection and analysis, decision to publish, or preparation of the manuscript.

==============================
Grafting is widely applied in the cultivation of melon. In this study, ‘Qinmi No.1’ (Cucumis melo L.(QG)) and ‘Ribenxuesong’ (Cucurbita maxima Duch. (RG)) were used as rootstocks for ‘Qingxin Yangjiaocui’ (Cucumis melo L.). The results showed that grafting with muskmelon rootstocks had no significant effect on fruit aroma, but grafting with pumpkin rootstocks significantly reduced the odor intensity and odor preference scores of melon fruits. Compared with the fruits from self-grafted plants (SG), four new aromatic volatiles with a sweet smell were detected, the alcohol dehydrogenase (ADH) activity was significantly decreased at 30 DAP, but unaffected at 42 DAP in QG fruits. There was no difference for alcohol acetyltransferase (AAT) activity between QG and SG fruits. The expression level of CmADH2 was significantly higher at 30 DAP and 42 DAP, but CmAAT2 was significantly lower at 42 DAP in QG fruits compared with SG fruits. In RG fruits, the main aroma compounds including butanoic acid ethyl ester, 2-methyl-2-butene-1-al, and 2-methylheptan-1-al were absent, while the volatile compounds with unpleasant odor characteristics including trans, cis-2,6-nonadien-1-ol, (E,E)-2,4-heptadienal, octanoic acid, and styrene were detected. Compared with SG fruits, 1-nonanol and 1-heptanol with green odor characteristics were significantly increased, but eucalyptol and farnesene with fruity aroma characteristics were significantly decreased in RG fruits. The ADH activity of RG fruits was significantly lower than that of SG fruits at 30 DAP and the AAT activity was significantly lower than that of SG fruits at 42 DAP. In addition, the expression levels of CmADH and CmAAT homologs in RG fruits were significantly lower than those in SG or QG fruits. These results show that grafting with pumpkin rootstocks affected the main aroma components, reduced ADH and AAT activities, and down-regulated the expression levels of CmADHs and CmAATs in the melon fruits. This study reveals the mechanism of different rootstocks on melon fruit aroma quality, and lays a theoretical foundation for the selection of rootstocks in melon production. Future studies using overexpression or CRISPR/CAS system to obtain stable transgenic lines of genes encoding key aromatic volatiles, would be promising to effectively improve the flavor quality of melon.

Introduction

Melon (Cucumis melo L.) is an economically important horticultural crop, and widely cultivated throughout the world (Condurso et al., 2012). In melon production, the plants are often subjected to a variety of biotic and abiotic stresses such as soil-borne diseases, drought, chill or heat which result in the decreased yield and fruit quality of melon (Colla et al., 2006; Davis et al., 2008; Kyriacou et al., 2017; Louws, Rivard & Kubota, 2010). Melon can be divided into thin-skinned and thick-skinned types according to agronomic characteristics (Liu et al., 2020). However, most of melon species, especially thin-skinned varieties, possess low resistance to adversity stress which results in a restricted application area. Fortunately, grafting is considered an important technique that is routinely practiced to enhance the resistance of multiple biotic and abiotic stresses, and improve the final yield and fruit quality in melon production (Edelstein et al., 2005). The sucrose contents and fruit weight were significantly increased by the application of several outstanding rootstocks on the melon ‘Proteo’ and ‘Galia’ (Colla et al., 2010; Soteriou, Papayiannis & Kyriacou, 2016). The fruits from grafted melon plants using ‘Tianzhen No. 1’ rootstock exhibited better organoleptic characteristics and higher soluble sugars content than that in non-grafted plants (Kaleem et al., 2022).

Traditionally, pumpkin rootstocks were commonly used for grafting melon on account of its strong stress resistance (Kyriacou et al., 2017). Nevertheless, the interaction between scion and rootstock during the grafting process can result in reduced survival of the grafted plants (Guan et al., 2015a). In addition, the flavor quality of the grafted plants using pumpkin rootstocks was frequently reduced and even lead to the production of bitter fruits (Zhang et al., 2019). Pumpkin rootstocks led to an obvious increase of organic acid content, but had no effect on the sugar content in melon fruits (Camalle et al., 2023). The soluble solid content of melon fruits in grafting plants with pumpkin rootstocks were significantly decreased, but the application of muskmelon rootstocks had no significant influence on the flavor and soluble solid content of melon fruits (Colla et al., 2006; Verzera et al., 2014; Zhang et al., 2019). Similarly, the bitter fruits were frequently occurred due to the application of unsuitable pumpkin rootstocks by changing the content of phospholipids, cucurbitacins, and flavonoids which were the key contributors for the occurrence of bitter fruits in ‘Balengcui’ melon (Zhang et al., 2019). Therefore, the selection and application of excellent muskmelon rootstocks were rapidly increasing in melon cultivation because the muskmelon rootstocks had almost no effect on the fruit quality of grafted plants (Guan, Zhao & Huber, 2015b; Rouphael et al., 2010). However, the reason for the influenced melon fruit quality resulting from grafting rootstock types was still unknown. The comprehensive comparison of volatile compounds in melon fruits from self-grafted plants, plants grafted onto pumpkin and muskmelon rootstocks were needed to further investigate the effect of rootstock types on melon fruit flavor quality of grafted plants.

The fruit quality of melon includes external quality and internal quality, among which flavor quality plays a leading role in consumption. The fruits of melon are valued for their aroma and sweetness (Kyriacou, Leskovar & Colla, 2018). Previous study showed that the sweetness of melon fruit was influenced by a wide range of rootstock and scion interactions (Tomić et al., 2022). Aroma reflects the interaction of a wide variety of volatile compounds and serves as an important indicator of the flavor quality of melon fruits (Beaulieu & Lancaster, 2007; Pang et al., 2012). To date, approximately 300 volatile aromatic compounds, including esters, alcohols, aldehydes, and ketones, have been identified in the fruits of different melon varieties (Kourkoutas, Elmore & Mottram, 2006; Nagashima et al., 2021; Obando-Ulloa et al., 2008; Spadafora et al., 2019; Shi et al., 2020). Most esters have a fruity and sweet taste, and act as the basic flavor substance in mature melon fruits (Lignou et al., 2014). For example, ethyl butyrate, methyl 2-methylbutyrate, and ethyl 2-methylpropionate were the key esters for producing satisfactory sweetness and fruity aromas in ‘cantalupensis’ and ‘reticulatus’ melon fruits (Condurso et al., 2012; Kourkoutas, Elmore & Mottram, 2006). Nevertheless, most aldehydes and alcohols such as (Z,Z)-3,6-nonadien-1-ol, 1-octanol, 1-pentanol, (E)-2-butenal, (E,Z)-2,6-nonadienal, (Z)-6-nonenal possess green, fresh, and cucumber flavors, and were described as cucumber-like, green and even foul melon fruits (Verzera et al., 2011). In inodorus honeydew melon, the key flavor compounds were C9 fatty aldehydes, especially (Z)-6-nonenal, (E)-2-nonenal, (Z,Z)-3,6-nonadien-1-ol, (E,Z)-2,6-nonadienal and 2-methylbutanal, which resulted in the occurrence of green and cucumber aroma fruits (Verzera et al., 2011).The quantity and proportion of different volatile compounds exhibited obvious differences among different melon varieties, which resulting in various flavor characteristics (Vallone et al., 2013). In grafting plants with pumpkin rootstocks ‘RS841’ and ‘P360’, the content of 1-pentanol, (Z) -3-hexen-1-ol, 1-Octanol, (E) -2-butenal, and geranyl acetate with acid, pungent, and green odors were significantly increased, and these volatile compounds had an adverse effect on the aroma of melon fruits (Condurso et al., 2012). However, the influence factors of different aromatic substances needed to be further researched. The mechanism that rootstock types affected fruit flavor quality by changing volatile compounds was still unknown.

Esters act as the main components of melon fruit aroma, and can be divided into straight chain esters and branched chain esters. The synthesis of straight chain esters mainly depends on the lipoxygenase (LOX) pathway. Unsaturated fatty acids are catalyzed by LOX and converted into C6 aldehydes which are synthesized into alcohols relying on the activity of alcohol dehydrogenase (ADH). Subsequently, alcohol acetyltransferase (AAT) participates in the acetylation of alcohols to produce esters (Tang et al., 2015; Zhang et al., 2017). The branched chain esters with aromatic rings are mainly synthesized by the amino acid pathway. Amino acids such as alanine, valine or leucine are converted into aldehydes under the action of aminotransferase and decarboxylase, and then aldehydes are transformed into esters by the action of ADH and AAT (Gonda et al., 2010). During melon fruit ripening, the expressions of CmADH1, CmADH2, CmAAT1 and CmAAT2 were steadily increased so that the activities of ADH and AAT were also significantly enhanced (El-Sharkawy et al., 2005; Yahyaoui et al., 2002). Therefore, the synthesis of ester precursor alcohols and acyl-CoA were promoted, and the esters were actively synthesized in the melon fruits (Chen et al., 2016). Interestingly, previous study showed that pumpkin rootstock grafting reduced the expression of CmADH1, CmADH2, CmAAT1 and CmAAT2 in melon fruit, but the results of different pumpkin rootstocks on the content of major esters and the activity of ADH and AAT in ripe fruit were inconsistent (Tian et al., 2012).

Although the effect of rootstock types on melon fruit flavor quality has been studied, the aroma components of the fruit are varying and complex due to varieties and environmental differences (Shao et al., 2022; Xiao et al., 2021). The effect of different rootstocks on various volatile components and overall fruit quality of melon are still unclear. ‘Qingxian Yangjiaocui’ (Cucumis melo L.), a famous local oriental melon variety in China, was renowned for its crisp sweet fruits and generally relied on grafting for cultivation and production because of its poor stress resistance. However, the fruits of the grafted plants especially the application of pumpkin rootstocks were always characterized by certain change in aroma. At present, the main volatile compounds contributing to the aroma of ‘Qingxian Yangjiaocui’ melon fruits were unknown, and the physiological and molecular mechanisms underlying the effects of different genotypes of rootstocks on melon fruit aroma have to be settled urgently.

In order to verify whether the change of melon flavor quality is due to the difference in the content of main aromatic substances, the odor difference of melon fruits produced by self-grafting plants and plants grafted on the rootstocks of ‘Ribenxuesong’ pumpkin and ‘Qinmi No. 1’ muskmelon was evaluated, and the volatile components of these fruits were determined and compared with the aid of headspace solid phase microextraction gas chromatography-mass spectrometry (HS-SPME-GC-MS). In addition, the activity and gene expression of alcohol dehydrogenase (ADH) and alcohol acyltransferase (AAT), the key enzymes in esters synthesis, were analyzed to comprehensively explore the physiological and molecular mechanism of the effect of different rootstocks on the flavor quality of melon fruit. These results will be helpful to the large-scale cultivation and popularization of ‘Qingxian Yangjiaocui’ melon.

Materials & Methods

Plant materials

The study was conducted between December 2019 and May 2020 in the greenhouse of the Dasima Modern Agricultural Garden, Baoding, Hebei Province, China. The oriental melon variety ‘Qingxian Yangjiaocui’ was used as a scion, while ‘Qinmi No. 1’ (Cucumis melo L.) and ‘Ribenxuesong’ (Cucurbita maxima Duch.) were used as rootstocks. Fruit from self-grafted plants of ‘Qingxian Yangjiaocui’ were used as a control and compared with fruits from grafted plants of ‘Qinmi No. 1’ and ‘Ribenxuesong’. Grafting was carried out on full opening of the first true leaves of the rootstocks and initial exposure of the first true leaves of the scion. The grafted seedlings were transplanted to the greenhouse 1 month post-grafting and arranged in randomized groups. Each experimental groups comprised three biological replicates of 20 plants, which were raised using standard management practices. Hand-pollinated bisexual flowers were marked, with three of the subsequently developing fruits being retained on each plant. At the end of the fruit expansion period (30 days after pollination, 30DAP) and the commercial ripening period (42 days after pollination, 42DAP), five plants were randomly selected from each plot, and 3 fruits of each plant were taken and crosscut. Thereafter, the central portions (2/5) of each fruit were extracted for determination of the activities and genes expression level of alcohol dehydrogenase (ADH) and alcohol acyltransferase (AAT). For the fruits of 42 DAP, odor scores were evaluated and volatiles were determined by HS-SPME-GC-MS (headspace solid-phase microextraction coupled with gas chromatography-mass spectrometry) (Fig. 1).

Figure 1 The SG, QG and RG fruits using for odor evaluation and HS-SPME-GC-MS.

Odor evaluation of mature fruit

Fruit odor evaluations were conducted in the laboratory of Hebei Agricultural University according to previous studies (Cozzolino et al., 2020). Twenty-seven judges were randomly selected and specially trained for two weeks by using melons samples purchased from different local markets. Each judge described the sensory quality through the sense of smell and the acceptability of the smell after tasting, and detected the melon samples six times in total. During each training process, new melon varieties were added to enrich the judge’s criteria for evaluating the aroma and preference of melon fruits. After determining the basic attributes and standards of fruit odor concentration and acceptable level, the given standards were divided into five levels corresponding to different odor descriptions, and the evaluation was quantified based on a score of 0–4. Therefore, the fruit odor intensity was ranked according to the odor intensity grade, where 0 = virtually odorless (water, control), 1 = a slight odor, 2 = a light odor, 3 = a strong odor, and 4 = a particularly strong odor. Subsequently, the evaluators were requested to rate their odor preferences using a 0 to 4 fruit odor preference scale, where 0 = unacceptable, 1 = unpleasant, 2 = acceptable, 3 = suitable, and 4 = particularly pleasant.

The mature fruits of self-grafted, ‘Qinmi No. 1’ and ‘Ribenxuesong’ grafted plants were cut into small pieces, marked with random letters, divided into three groups and placed in three independent compartments of the laboratory. Nine judges formed an evaluation team, and each evaluation team entered an independent compartment. The team members evaluated the odor intensity and odor preferences of the mature fruits of three grafted plants through smell and taste, and scored them. After each taste, they rinsed their mouths with clear water. The experiment was carried out simultaneously in the compartment as three replicates of the odor evaluation test.

Metabolite extraction and gas chromatography–mass spectrometry analysis

To identify volatiles in melon fruits, the HS-SPME-GC-MS technology was performed (Baena-Pedroza, Londoño Giraldo & Taborda-Ocampo, 2020; Lasekan et al., 2013; Ma et al., 2018). Firstly, volatile compounds from melon fruits were collected using headspace solid-phase microextraction technology. Samples (300 mg) of fresh pulp homogenate containing 10 µL of 10 mg⋅L−1 2-octanol solution (added as an internal standard) were placed in 20 mL headspace bottles. A 50/30um DVB/CAR/PDMS fiber (Supelco, PA, USA) was exposed to the headspace of the sample for 30 min at 60 °C for SPME analysis. Subsequently, GC-MS was performed to obtain the original peak maps of each sample. Following extraction, the SPME device was manually inserted into the split/split-less inlet of the GC-MS system (Agilent 7890b/5977b) and held in split-less mode at 250 °C to desorb aromatic volatile compounds over a 4-min period. The volatile components were separated on a DB-Wax chromatographic column (30 m ×0.25 mm × 0.25 µm). Helium was used as the carrier gas with a front inlet purge flow rate of three mL min−1 and the flow rate through the column was one mL min−1. The initial GC oven temperature was set to 40 °C, at which it was held for 4 min. Thereafter, the temperature was increased to 245 °C at a rate 5 °C min−1 and held for 5 min. The MS transmission line, ion source, and quadrupole temperatures were maintained at 250 °C, 230 °C, and 150 °C, respectively. The mass spectrometer was operated in electron impact mode at −70 eV, with a scan range from m/z 20 to 400 and solvent delay of 0 min.

Chroma TOF 4.3X software of LECO Corporation and National Institute of Standards and Technology (NIST, https://www.nist.gov/srd) database were used for raw peaks exacting, the data baselines filtering and calibration of the baseline, peak alignment, deconvolution analysis, peak identification, integration and spectrum match of the peak area. According to the retention time, retention index and mass spectrum information, it is matched with NIST database to characterize the detected volatile components (Wang et al., 2022). The relative content of volatiles was evaluated based on the ratio of the peak area of 2-octanol to the peak area of detected volatiles (Table S1) (Kind et al., 2009).

Detection of alcohol dehydrogenase and alcohol acyltransferase enzyme activities

Alcohol dehydrogenase (ADH) activity was determined by ethanol dehydrogenase detection kit (Shanghai Yuanye, China). The activity of alcohol acyltransferase (AAT) in fruits was determined according to (Chen et al., 2016). The frozen samples (3 g) of melon fruits were ground in liquid nitrogen, and 2.25 mL of enzyme extracting solution was added to continually extract for 20 min on ice, and after that, the samples were centrifuged at 12,000 g for 20 min under 4 °C. The supernatant was taken as the crude enzyme solution for the AAT enzyme activity assay. The crude enzyme solution (0.6 mL) was reacted with 2.5 mL of reaction solution (pH = 8.0). The reaction solution contains 5 mM Tris–HCl buffer MgCl2, 0.15 ml 0.5 mM acetyl-CoA and 0.05 ml 200 mM butanol. After reacting at 30 °C for 15 min, 0.1 mL of 10 mM 5,5-disulphide dinitrobenzoic acid (DTNB) was added to the mixed solution and left at room temperature for 10 min. Colourimetry was carried out with a UV spectrophotometer at 412 nm and each sample was repeated three times.

Real-time quantitative RT-PCR

The total RNA of grafted fruit was extracted by rapid universal plant RNA isolation Kit (Huayueyang, China), and the cDNA was obtained by reverse transcription amplification of total RNA by FastQuant cDNA Kit (Tiangen Biotech, Beijing, China). SYBR Green PCR Master Mix (Roche Diagnostics, Basel, Switzerland) was used for quantitative real-time RT-PCR on Roche lightcycle 96 real-time PCR system (Applied Biosystems, Waltham, MA, USA). The gene specific primers of RT-PCR are shown in Table 1. The β-actin was used as an internal reference control gene to standardize gene expression data. The 2−ΔΔCt method was used to calculate the relative gene expression of key regulatory enzymes of volatile substances in grafted plants.

Table 1 Sequence of primers used for gene expression analysis by real-time quantitative PCR.

Name	Oligonucleotide sequence	
	Forward primer sequence (5′–3′)	Reverse primer sequence (5′–3′)	
CmAAT1	CCACAGGGGCCAGAATTAC	TGGAGGAGGCAAGCATAGACT	
CmAAT2	CTATAATTGGAGGGTGTGGAATTATC	AACATTTGCCCTAAATCTTTCCAT	
CmADH1	GTGTTCTTAGCTGCGGCATTT	TTGACCCTTTTTAGGCTTTGCA	
CmADH2	GCGGAATCGTTAAAGGGTGTA	AGCCGCCTCTCTCTCTTCTTC	
β-actin	CCGTTCTGTCCCTCTATGCT	AGTAAGGTCACGACCAGCAA	
Notes.

Gene registration number is from Melonomics database v4.0 (https://www.melonomics.net/). β-actin: MELO3C023264; CmAAT1: MELO3C024766; CmAAT2: MELO3C024771; CmADH1: MELO3C003251; CmADH2: MELO3C027151. The relative expression of CmADH1, CmADH2, CmAAT1 and CmAAT2 relative to SG fruits at 30 DAP (control) were detected in RG and QG fruits.

Statistical analysis

Principal component analysis, orthogonal-partial least squares discriminant analysis (OPLS-DA), and multivariate statistical analysis of the GC-MS data were performed using R software (Supplementary File 1). Other data were analyzed with an analysis of variance (ANOVA) using the SPSS 25.0 statistical package (SPSS Inc., Chicago, IL, USA). For each experiment, significant differences were determined based on a one-way ANOVA and Duncan’s multiple range test at the p < 0.05 level.

Results

Evaluation of the odors of mature fruits from self-grafted plants and plants with different rootstocks

In order to study the effect of different rootstocks on overall fruit quality of melon, the odor scores including odor intensity and odor preferences of melon fruits produced by self-grafted plants (SG) and plants grafted onto muskmelon (‘Qinmi No. 1’, QG) and pumpkin (‘Ribenxuesong’, RG) rootstocks were evaluated (Table 2). The melon fruits grafted onto ‘Qinmi No. 1’ rootstock possessed the highest odor intensity of 3.67, followed by the fruits of self-grafted plants (3.22), and the fruit odor intensity causing by the ‘Ribenxuesong’ rootstock was the lowest (2.78). The odor intensity of QG fruits was significantly higher than that in RG fruits, but there was no significant difference in odor intensity between SG and RG fruits (Table 2). The odor preference score of SG fruit was 3.44. Compared with SG fruits, the odor preference scores of QG (3.33) and RG (2.4) fruits were decreased, but there was no significant difference in odor preference between QG and SG fruits. Both the odor preference scores of SG and QG fruits were significantly higher than that in RG fruits (Table 2). However, no obvious changes in external quality such as fruit shapes and color were observed after grafting using different rootstocks (Fig. 1). These results indicated that rootstocks mainly led to the odor difference of ‘Qingxian Yangjiaocui’ melon fruits.

Table 2 Intensities and preferences for the odors of melon fruits produced by self-grafted plants (SG) and plants grafted onto ‘Qinmi No. 1’ (QG) and ‘Ribenxuesong’ (RG) rootstocks.

Treatment	SG	QG	RG	
Odor intensitya	3.22 ab	3.67 a	2.78 b	
Odor preferencesb	3.44 a	3.33 a	2.4 b	
Notes.

Different letters in the same row represent significant differences at the P < 0.05 level, as determined using Duncan’s multiple range test.

a Melon fruit odor intensity (0 = virtually odorless, 1 = slight odor, 2 = light odor, 3 = strong odor, 4 = particularly strong odor).

b Melon fruit odor preference (0 = unacceptable, 1 = unpleasant, 2 = acceptable, 3 = suitable, 4 = particularly pleasant).

Identification and analysis of volatile compounds in mature fruits from self-grafted plants and plants with different rootstocks

To investigate the reason for the odor change, the volatile compounds in melon fruits of SG, QG and RG were determined by HS-SPME-GC-MS technology (Table S2). The total ion flow chromatograms obtained based on GC-MS analyses of volatiles in the fruits of plants grafted onto different rootstocks revealed good sample separation and peak overlap in parallel tests, thereby indicating the good reproducibility of sample analyses (Fig. 2). The total number of volatiles detected in SG, QG and RG fruits was 211, 211 and 216 respectively, and the corresponding relative contents were 6,193.05, 7,043.62 and 6678.35µg⋅L−1 respectively. These compounds included esters, alcohols, hydrocarbons, aldehydes, ketones, nitrogen compounds, phenolics, acids, and sulfur compounds (Table S2). To gain further insights into the influence of rootstocks on melon fruit volatile composition, principal component analysis was performed on all volatiles detected, the resulting ordination plots of which indicated that whereas there was a notable overlap of volatiles in QG and SG fruits, RG and SG fruits were clearly separated within 95% confidence intervals (Fig. 3, Table S3). These findings accordingly revealed a high similarity between the volatile profiles of QG and SG fruits and significant difference between the volatiles in RG and SG fruits.

Figure 2 Total ion current chromatograms of volatile compounds in melon fruits produced by plants grafted onto different rootstocks.

SG, self-grafted; QG, grafted on a ‘Qinmi No. 1’ rootstock; RG, grafted on a ‘Ribenxuesong’ rootstock. In all cases, ‘Qingxian Yangjiaocui’ melon was used as the scion. The values in the figure refer to the matched compounds. 1: 2-octanol (internal standard); 2: 2-methylheptan-1-al; 3: butanoic acid ethyl ester; 4: 2-methyl-2-butene-1-al; 5: ethyl 3-methylbutyrate; 6: ethyl 3-hydroxybutyrate; 7: ethyl 3-(methylthio) propionate; 8: 1,4-nonanolactone; 9: styrene; 10: (E, E)-2,4-Heptadienal; 11: citral; 12: trans, cis-2,6-nonadiene-1-ol; 13: octanoic acid.

Figure 3 Principal component analysis of the metabolite profiles of volatile compounds in fruits produced by plants grafted onto different rootstocks.

SG: self-grafted; QG: grafted on a ‘Qinmi No. 1’ rootstock; RG: grafted on a ‘Ribenxuesong’ rootstock.

Volatile compounds in the mature fruits of self-grafted plants

As shown in Table 3, the contents of esters, alcohols, hydrocarbons, and aldehydes in SG fruits accounted for 87.39% of the total content of volatile compounds. With 44 identified species, esters comprised the largest category of volatiles detected in SG fruits with a proportion of 26.4%. Among these, 2,3-butanediol diacetate, 2-methyl-1-butanoate, butyl acetate, isobutyl acetate, ethyl acetate (methylthio) ester, 1,3-butanediol diacetate, ethyl butyrate, dibutyl phthalate, dl-pantothenic acid lactone, ethyl 2-methylbutyrate, 1-octene-3-acetate, ethyl propionate, and hexyl acetate were identified as the predominant types, accounting for 93.34% in all esters (Table S2). The alcohol content was slightly lower, and 37 alcohols were identified which accounted for 46.72% of the total volatile content in SG fruits (Table 3). Among these, dimethylsilanediol, 1-octen-3-ol, 1-hexanol, 2-methyIthioethanol, 1-nonanol, benzyl alcohol, and 3-(methylthio)-1-propanol accounted for 81.4% in the total alcohols (Table S2). In total, 19 aldehydes were identified in SG fruits, and their relative contents were 4.23% in the total volatiles, among which, heptanal, benzaldehyde, (Z)-6-nonenal, 2-methyl-2-butene-1-al, (E, E)-2,6-nonadienal, 2-methylheptan-1-al, nonanal, benzene acetaldehyde, and hexanal accounted for 86.1% of the total aldehyde content (Table 3; Table S2). In addition, 43 hydrocarbons were detected with a proportion of 10.01% in the total volatiles of SG fruits (Table 3). Unlike the esters with a fruity and sweet taste, alcohols, and aldehydes, hydrocarbons were odorless compounds, and assumed to make little contribution to the aroma of ‘Qingxian Yangjiaocui’ melon fruits.

Table 3 Types and relative contents of volatile compounds in melon fruits produced by self-grafted plants (SG) and plants grafted onto ‘Qinmi No. 1’ (QG) and ‘Ribenxuesong’ (RG) rootstocks.

Caegories	SG	QG	RG	
	Number of compounds	Relative contents (µg⋅L−1)	Number of compounds	Relative contents (µg⋅L−1)	Number of compounds	Relative contents (µg⋅L−1)	
Esters	44	1,636.8	47	1,350.83	47	1,222.91	
Alcohols	37	2,893.23	34	3,233.50	40	3,394.78	
Hydrocarbones	43	619.98	41	1,014.56	40	994.09	
Aldehydes	19	261.69	17	323.64	18	233.54	
Ketones	20	189.58	18	185.7	21	212.94	
Nitrogen compounds	8	53.43	12	47.05	10	85.87	
Phenolics	8	28.84	8	21.32	9	47.95	
Acids	4	24.59	5	7.32	7	32.39	
Sulfur compounds	3	8.23	3	7.86	1	4.62	
Others	25	476.69	26	851.86	23	449.26	
Total	211	6,193.05	211	7,043.62	216	6,678.35	

Different volatile compounds existed in the mature fruits of self-grafted plants and grafted plants with pumpkin or muskmelon rootstocks

In order to explore the effects of different rootstocks genotypes on melon fruits aroma, the volatile compounds of QG and RG fruits were respectively compared with that of SG fruits (Control). In QG fruits, a total of 47 esters, 34 alcohols and 17 aldehydes were detected (Table 3). Among these, 28 volatiles were found to different from that detected in SG fruits (six esters, two alcohols, three hydrocarbons, three ketones, five nitrogenous compounds, three acids, one phenol, and five other compounds), which collectively accounted for 0.85% in the total volatile components in QG fruits (Table 4). According to the description of flavornet and human odor space (http://www.flavornet.org/index.html), four of these volatiles, namely, ethyl 3-(methylthio) propionate (fruity flavor), ethyl 3-hydroxybutyrate (marshmallow flavor), ethyl 3-methylbutyrate (fruity flavor), and 1,4-nonanolactone (coconut and peach flavor), with relative contents ranging from 0.4 to 5.95 µg⋅L−1, were described as the sweet aroma. In addition, among the volatiles identified in SG fruits, 28 compounds were not detected in QG fruits, which included five esters, five alcohols, five hydrocarbons, four ketones, two aldehydes, one nitrogenous compound, one phenol, two acids, and three other compounds that accounted for 1.2% in the total volatiles of SG fruits (Table 4). However, no specific odor descriptions were found for these compounds and they were not the main aroma components in SG fruits. Therefore, the important aromatic volatiles were reserved in QG fruits, and four new aromatic components were produced compared with SG fruits.

Table 4 Relative contents (µg⋅L−1) of specific volatile compounds in melon fruits produced by self-grafted plants (SG) and plants grafted onto ‘Qinmi No. 1’ (QG) and ‘Ribenxuesong’ (RG) rootstocks.

Compounds	Relative contents (µg⋅L−1 )	
	SG	QG	RG	
Esters				
Butanoic acid ethyl ester	36.30	46.23	nda	
Acetic acid diphenyl-hydroxy-1-dimethylaminoisopropyl ester	7.93	nd	nd	
Diethyl 2-(N-(tert-butoxycarbonyl)amino)malonate	3.14	nd	0.69	
2,2′-Ethylenedioxydi-ethanodiacetate	1.98	nd	nd	
Citronellyl butyrate	1.74	1.77	nd	
2-Ethylhexyl salicylate	0.37	nd	0.35	
Isophthalic acid di(2-isopropylphenyl) ester	0.18	nd	0.12	
Isobutyl 2-methylcrotonate	nd	nd	6.42	
1,4-Nonanolactone	nd	3.49	2.24	
Heptanoic acid 3-nitrophenyl ester	nd	nd	1.23	
2-Benzofurancarboxylic acid, 2,4,5,6,7,7a-hexahydro-4,4,7a-trimethyl-, methyl ester, cis-	nd	nd	1.13	
1,3-Dioxolane-4-methanol, 2-pentadecyl-, acetate, trans-	nd	0.45	0.75	
Acetyl eugenol	nd	nd	0.41	
3,7-Dimethyl-6-octenyl 3-methylbutanoate	nd	nd	0.26	
Ethyl 3-methylbutyrate	nd	0.4	nd	
Ethyl 3-hydroxybutyrate	nd	1.93	nd	
Ethyl 3-(methylthio) propionate	nd	5.95	nd	
Fumaric acid, di(cis-non-3-enyl) ester	nd	0.26	nd	
Carbonic acid decyl undecyl ester	nd	1.58	nd	
Ethyl 9-hexadecenoate	nd	0.15	nd	
Alcohols				
Trans-1,4-dihydroxycyclohexane	1.25	nd	nd	
O-(2-chloropropionyl)-O’-(4-fluorobenzoyl)- 1,2-benzendiol,	1.03	nd	0.79	
(3R,3aS,6S,7R)-3,6,8,8-Tetramethyloctahydro-1H-3a,7-methanoazulen-6-ol	0.32	nd	0.33	
1-Ethynyl cyclohexanol,	0.2	nd	nd	
1-(2-Furyl)-3-butene-1,2-diol	0.07	nd	nd	
trans,cis-2,6-Nonadien-1-ol	nd	nd	1.57	
2-Buten-1-ol, (Z)-	nd	nd	1.13	
2-Cyclohexen-1-ol, 2,4,4-trimethyl-	nd	nd	0.9	
Silanol, ethyldimethyl-	nd	nd	0.85	
2-Cyclohexen-1-one, 4-(3-hydroxy-1-butenyl)-3,5,5-trimethyl-	nd	0.69	0.6	
Bicyclo[3.3.1]nonan-3-ol, exo-	nd	nd	0.36	
16-Methyl-heptadecane-1,2-diol, trimethylsilyl ether	nd	0.91	nd	
Hydrocarbones				
4,5-Dimethyl-2,6-octadiene.	nd	6.31	9.85	
2,4-Dimethylheptane	0.91	nd	1.58	
Hexadecane	0.83	nd	1	
Tricyclo[4.3.1.0(2,5)]decane	nd	nd	0.82	
2-Methyladamantane	0.62	nd	0.41	
Megastigma-4,6(Z),8(E)-triene	1.4	1.07	nd	
Cyclohexylmethyldimethoxysilan	0.73	1.01	nd	
1-Hexadecyne	nd	0.4	nd	
2,6,10-Trimethyldodecane,	1.27	0.38	nd	
Nonylcyclopentane	nd	0.3	nd	
4-Methyloctane	1.44	nd	nd	
1-Methyl-2-methylenecyclopentane	1.18	nd	nd	
Aldehydes				
2-Methyl-2-butene-1-al	22.64	27.42	nd	
2-methylheptan-1-al	15.81	16.59	nd	
4-Heptenal	4.26	nd	nd	
2,3-dihydro-1H-Indene-4-carboxaldehyde	0.49	nd	nd	
(E,E)-2,4-Heptadienal	nd	nd	5.02	
Citral	nd	nd	1.36	
5-Ethylcyclopent-1-enecarboxaldehyde	nd	nd	0.9	
Ketones				
1,3-Cyclopentanedione, 4-(3-methylbutyl)-	nd	nd	0.69	
4-Methyl-5-nonanone	0.59	nd	nd	
1-Hepten-3-one	nd	1.5	2.57	
Isophorone	2.67	nd	0.93	
4a,8a-(Methaniminomethano)naphthalene-9,11-dione, 10-phenyl-	0.47	nd	nd	
(8Z)-1-oxacycloheptadec-8-en-2-one	nd	nd	5.01	
Nitrogen compounds				
1,2-Benzenedicarbonitrile	0.48	0.27	nd	
Aminoacetonitrile	nd	0.52	nd	
2-Butyl-1-methylpyrrolidine	nd	4.8	nd	
(2-Hydroxyethyl)trimethylammonium bromide	nd	8.24	nd	
N-(phenylmethyl)-acetamid	nd	nd	0.22	
Caprolactam	nd	0.3	0.41	
Semustine	nd	3.14	3.94	
Dothiepin	27.52	nd	46.54	
Acids				
3-Octenoic acid, TMS derivative	17.43	nd	21.73	
Octanoic acid	nd	nd	2.21	
Palmitoleic acid	1.13	nd	1.06	
3,3-Dimethylbutyl propylphosphonofluoridate	nd	0.09	nd	
(Z,Z)-Octadeca-9, 12-dienoic acid	nd	0.5	0.6	
n-Hexadecanoic acid	nd	1.12	2.29	
Phenolics				
2-Methoxy-5-methylphenol	nd	nd	5.68	
2,6-Dimethoxy-4-prop-2-enylphenol	0.28	nd	0.26	
3,5-Bis(1,1-dimethylethyl)-2-benzenediol	nd	1.27	nd	
Sulfur compounds				
Carbon disulfide	1.14	1.4	nd	
2-(1,1-Dimethylethoxy)thiophene	2.21	2.14	nd	
other				
2-Methyl-2-(5-phenyl-3-pentenyl)-1,3-dioxolane	nd	13.09	8.37	
Styrene	nd	2.14	5.65	
Benzene	1.39	nd	1.76	
1,2-Dimethoxy-4-(1-propenyl)-benzen	nd	nd	0.18	
2-Pentoxy-tetrahydropyran	nd	nd	0.06	
Indole, 3-methyl-2-(2-dimethylaminopropyl)-	13.03	18.94	nd	
Methyl 2,3,4,6,7-penta-O-methyl-L-glycero-D-mannoheptopyranoside	0.56	0.98	nd	
Butyl aldoxime, 2-methyl-, syn-	nd	0.97	nd	
2-Propenenitrile, 3-phenyl-, (E)-	0.39	0.79	nd	
3-Methyl-2-(3,7,11-trimethyldodecyl) furan	nd	0.64	nd	
2-Pentadecyl-1,3-dioxolane	nd	0.51	nd	
Trifluoromethyldifluorophosphine	0.19	nd	nd	
2,4,4,6-Tetramethyl-6-phenylheptane	0.57	nd	nd	
3-Butylisobenzofuran-1(3H)-one	0.6	nd	nd	
Notes.

a nd, was not detected.

In RG fruits, 47 esters, 40 alcohols and 18 aldehydes were detected (Table 3). Among all detected volatiles, 32 compounds were not found in SG fruits, including seven esters, six alcohols, three aldehydes, two hydrocarbons, three ketones, three nitrogenous compounds, three acids, one phenolic compound, and four other compounds, which collectively accounted for 1.1% of the total volatile compounds in RG fruits (Table 4). Among these volatiles, 1,4-nonanolactone (0.41 µg⋅L−1) was described as the coconut and peach flavor; citral (1.36 µg⋅L−1) and trans, cis-2,6-nonadiene-1-ol (1.57 µg⋅L−1) were characterized by the lemon and cucumber flavor, respectively; (E,E)-2,4-Heptadienal (5.02 µg⋅L−1) had a nutty flavor; octanoic acid (2.21 µg⋅L−1) possessed a sweaty smell; and styrene (5.65 µg⋅L−1) had a distinct gasoline-like smell. Conversely, there were 27 compounds present in SG fruits that were not detected in RG fruits, namely, four esters, three alcohols, five hydrocarbons, three ketones, four aldehydes, one nitrogenous compound, two sulfur-containing compounds, and 5 other compounds, the total contents of these 27 compounds accounted for 1.8% of all volatiles in SG fruits (Table 4). Among them, butanoic acid ethyl ester, 2-methyl-2-butene-1-al, and 2-methylheptan-1-al were determined to be the major volatiles in SG fruits at 36.3, 22.64 and 15.8 µg⋅L−1, respectively (Table 4), butanoic acid ethyl ester and 2-methyl-2-butene-1-al were described as the fruity aroma. These results suggested that RG fruits produced some new aromatic volatiles, but the main aromatic volatiles were absent compared to SG fruits.

Differences in the contents of volatile compounds in mature fruits of self-grafted and grafted plants with pumpkin or muskmelon rootstocks

OPLS-DA analysis revealed that only four compounds were assigned variable importance factor (VIP) values >1, the contents of which in QG and SG fruits were found to significantly differ (P < 0.05) (Table 5). In these four compounds, the relative contents of the hydrocarbon hexamethyl-cyclotrisiloxan and the phenolic compound 1-ethylphenol were significantly higher in QG fruits than that in SG fruits. On the contrary, the relative contents of the alcohol eucalyptol and the hydrocarbon silane dimethyl(dimethyl(dimethyl(2-isopropylphenoxy)silyloxy) silyloxy) (2-isopropylphenoxy)-silane were significantly reduced in QG fruits. However, there was no significant difference in the levels of the main volatiles in SG and QG fruits because these compounds were not the main volatiles in ‘Qingxian Yangjiaocui’ melon fruits.

Table 5 Differential volatiles in melon fruits of self-grafted plants (SG) and plants grafted onto a ‘Qinmi No. 1’ (QG) rootstock.

Compounds	VIP	Ratio a	
Hydrocarbones			
Dimethyl(dimethyl(dimethyl(2-isopropylphenoxy)silyloxy)silyloxy)(2-isopropylphenoxy) silane	1.9124	0.8625	
Hexamethyl-cyclotrisiloxan	2.2151	4.1438	
Alcohols			
Eucalyptol	2.2338	0.7609	
Phenolics			
2-Ethylphenol	2.173	2.285	
Notes.

a The ratio of the relative contents of volatile components in the fruits of plants grafted on ‘Qinmi No. 1’(QG) to those of self-grafted plants (SG).

Table 6 presents a comparison of the volatiles detected in RG and SG fruits, among which 24 compounds had VIP values >1 and their relative contents were significantly different between RG and SG fruits (P < 0.05). The relative contents of 14 compounds were significantly higher in RG fruits than that in SG fruits, although the odors of only four compounds had been described. 2-methyl-1-butanoacetate and 2-pentylfuran had a sweet aroma, and 1-nonanol and 1-heptanol possessed a grass odor. The remaining 10 differential compounds were significantly decreased in RG fruits, among which, only the odors of eucalyptol and farnesene have been previously described to have a sweet aroma.

Table 6 Differential volatiles in melon fruits produced by self-grafted plants (SG) and plants grafted onto a ‘Ribenxuesong’ (RG) rootstock.

Compounds	VIP	Ratio a	
Esters			
Ethyl 4-(ethyloxy)-2-oxobut-3-enoate	1.758	1.992	
2-methyl-1-butanoacetate	1.671	1.325	
Diethyl 2-(N-(tert-butoxycarbonyl)amino)malonate	1.537	0.219	
Alcohols			
1,1-Dimethoxy-2-propanol	1.835	13.31	
(6Z)-Nonen-1-ol	1.839	3.303	
1-Heptanol	1.601	2.1	
1-Nonanol	1.939	1.946	
Eucalyptol	2.018	0.651	
1-(2-Furyl)-3-butene-1,2-diol	1.609	0.193	
Hydrocarbones			
Hexamethyl-cyclotrisiloxane	1.949	3.941	
2,6-Dimethyl-2-octene	1.585	1.904	
1,1,1,5,5,5-Hexamethyl-3,3-bis[(trimethylsilyl)oxy]-Trisiloxane	1.87	1.22	
Phenyl-pentamethyl-disiloxane	1.811	0.88	
Farnesene	1.617	0.586	
Ketones			
6-Octen-2-one	1.135	0.363	
4-Methyl-5-nonanone	1.553	0.227	
Aldehydes			
cis,cis-7,10,-Hexadecadienal	1.856	0.674	
2,3-dihydro-1H-Indene-4-carboxaldehyde	1.227	0.308	
Nitrogen compounds			
5H-Tetrazol-5-amine	1.718	1.637	
Others			
1,2-Dimethoxy-4-(1-propenyl) benzene	1.641	5.851	
2,2′-Trimethylenebis-1,3-dioxolane	1.915	3.487	
2-Pentylfuran	1.801	2.401	
Furan, 3-(4-methyl-3-pentenyl)-	1.825	0.525	
Trifluoromethyldifluorophosphine	1.578	0.213	
Notes.

a The ratio of the relative contents of volatile components in the fruits of plants grafted on ‘Ribenxuesong’ (RG) to those of self-grafted plants (SG).

Differences in activities of ADH and AAT in SG, QG and RG fruits

The ADH activity of ‘Qingxian Yangjiaocui’ fruit at 30 DAP was higher than that at 42 DAP (Fig. 4A). At this time, the ADH activity in QG and RG fruits was significantly lower than that in SG fruits, and the ADH activity in RG fruits was also significantly decreased compared with QG fruits. At 42 DAP, the ADH activity in fruit decreased by 39.4%-86.0% compared with that at 30 DAP, there were no significant differences in ADH activity among QG, RG and SG fruits (Fig. 4A, Table S4).

Figure 4 Effects of rootstocks of different genotypes on ADH (A) and AAT (B) activities of melon fruit.

Error bars represent ± SD. SG, self-grafted; QG, grafted on a ‘Qinmi No. 1’ rootstock; RG, grafted on a ‘Ribenxuesong’ rootstock. Different letters indicate significant differences in statistics in the same period (p < 0.05).

The AAT activity was extremely low at 30 DAP in SG fruits (Fig. 4B). There were no significant differences in AAT activity among QG, RG and SG fruits at 30 DAP. At 42 DAP, AAT activity reached 4.8−8.1 times compared with that at 30 DAP. The AAT activity in QG fruits had no difference from that in SG fruits, but the AAT activity in RG fruits were significantly lower than that in SG and QG fruits (Fig. 4B, Table S4).

Differences in gene expression of CmADH and CmAAT in SG, QG and RG fruits

At the stage of 30 DAP and 42 DAP, the expression level of CmADH1 had no difference between QG and SG fruits, but it was significantly decreased in RG fruits (Fig. 5A). While the expression level of CmADH2 in QG fruits was significantly higher than that in SG fruits, but it was significantly reduced in RG fruits (Fig. 5B). At the stage of 30 DAP, there was no significant difference for expression level of CmAAT1 and CmAAT2 in SG, QG and SG fruits. At 42 DAP, the expression level of CmAAT1 in QG fruits was also not different from that in SG fruits, but CmAAT2 was significantly decreased in QG fruits compared with SG fruits. However, the expression levels of CmAAT1 and CmAAT2 in RG fruits were significantly lower than that in SG and QG fruits (Figs. 5C, 5D, Table S5).

Figure 5 Effects of rootstocks of different genotypes on the expression of alcohol dehydrogenase and alcohol acyltransferase-related genes in mature melon fruit.

Different letters indicate significant differences in statistics in the same period (p < 0.05). Error bars represent ± SD. SG, self-grafted; QG, grafted on a ‘Qinmi No. 1’ rootstock; RG, grafted on a ‘Ribenxuesong’ rootstock. Alcohol dehydrogenase CmADH1 (A). Alcohol dehydrogenase CmADH2 (B). Alcohol acyltransferase CmAAT1 (C). Alcohol acyltransferase CmAAT2 (D).

Discussion

Esters play an important role in determining the aroma of ‘Qingxian Yangjiaocui’ melon fruits

Melons are typically classified into aromatic and non-aromatic types based on their fruit aroma intensities which are attributable to the combined effects of a diverse array of volatile compounds (Esteras et al., 2018; Mayobre et al., 2021). It has previously been established that different types of melon fruit can differ significantly with respect to the compositions and contents of these volatile compounds (Chaparro-Torres, Bueso & Fernández-Trujillo, 2016; Fredes et al., 2016). For example, aromatic melon fruits are rich in esters, among which, ethyl acetate, butyl acetate, isobutyl acetate, hexyl acetate, pentyl acetate, benzoyl acetate, 2-methylbutyl acetate, ethyl 2-methylbutyrate, ethyl hexanoate, and ethyl butyrate have been identified as the main ester components (Beaulieu & Lancaster, 2007; Kende et al., 2019; Tang et al., 2015). In contrast, the volatiles of non-aromatic melon fruits typically comprise a larger proportion of alcohols and aldehydes, which account for a proportion of 75.0% to 98.1% in total fruit volatiles, whereas the proportion of esters is only 0.7% to 11.7% in total (Dos-Santos, Bueso & Fernández-Trujillo, 2013; Verzera et al., 2011). In non-aromatic melon fruits, the contents of (Z)-6-nonenal, 1-nonanal, trans, cis-2,6-nonadienal, trans-2-nonenal, decaldehyde, and trans-2, cis-6-nonadienol, which contribute to the fresh cucumber flavor of fruit, have been found to be considerably higher than corresponding levels in the fruits of aromatic melon (Buescher & Buescher, 2001; Perry, Wang & Lin, 2009).

The fruits of ‘Qingxian Yangjiaocui’ have a particularly distinct aroma, but there were no studies reporting the main volatile compounds contributing to the aroma of ‘Qingxian Yangjiaocui’ melon fruits. In this study, esters are the most diverse and abundant volatiles in the fruits of self-grafted plants, in which 13 main esters components accounting for 93.34% of total esters (Table S2). Previous study reported that 2-methyl-1-butanoacetate, acetic acid butyl ester, isobutyl acetate, butanoic acid ethyl ester, 2-methylbutanoic acid ethyl ester, propanoic acid ethyl ester, and acetic acid hexyl ester were typical aroma volatiles of aromatic melon fruits (Pang et al., 2012). In addition, seven alcohols and nine aldehydes were identified to affect the aroma of ‘Qingxian Yangjiaocui’ fruits (Table S2). Therefore, the aroma of ‘Qingxian Yangjiaocui’ melon fruits was mainly due to esters, but also influenced by a combined effects including alcohols and aldehydes.

Grafting with muskmelon rootstocks was beneficial to improve the flavor quality of melon

Grafting was commonly practiced in melon cultivation, but the choice of rootstocks may have a notable influence on the aroma of fruits produced by grafted plants (Tripodi et al., 2020). In this study, there was no significant difference in the perceived sensory characteristics between self-grafted (SG) fruits and those produced by plants grafted on muskmelon rootstocks (QG) (Table 2). Volatile compounds play important roles in determining melon flavor, and rootstocks have been demonstrated to influence fruit aromas by affecting volatile compounds. Condurso et al. (2012) reported that muskmelon rootstocks had no significant effects on esters in melon fruits, or even enhanced fruit ester contents. However, muskmelon rootstock ‘PG22HF1’ reduced the content of aromatic esters such as ethyl acetate, hexyl acetate, phenylmethyl acetate, and 2-methylbutyl acetate in ‘Yumeiren’ melon fruits, and the expression levels of CmADH1, CmADH2, CmAAT1, and CmAAT2 in fruits were also significantly reduced, but the activities of ADH and AAT in fruits were not significantly affected (Tian et al., 2012).

In this study, volatiles in the fruits of QG and SG were highly similar in terms of both composition and relative contents, and the fruits of QG produced four new volatiles with sweet smell, including ethyl 3-(methylthio)propionate, ethyl 3-hydroxybutyrate, ethyl 3-methylbutyrate, and 1,4-nolactone. The detection of four new aromatic volatiles with a sweet smell in QG fruits compared to SG fruits highlights the potential benefits of grafting in enhancing aroma. The activity of ADH in QG fruits was significantly lower than that of SG fruits only at 30 DAP, while the AAT activity of the QG fruits has no difference with that of SG fruits (Fig. 4). In addition, there was no difference in the expression levels of CmADH1 and CmAAT1 between QG fruit and SG fruits at 30 DAP and 42 DAP, while the expression level of CmADH2 was significantly higher than those of SG fruits, and the expression level of CmAAT2 was significantly lower than those of SG fruits only at 42 DAP (Fig. 5). Besides AAT and ADH, added regulators contributing to different aromatic substances by grafting need to be further investigated.

Grafting with pumpkin rootstocks has a negative effect on melon flavor quality

Pumpkin rootstock grafting reduces the content of key aroma volatiles in ripe fruit, but individual pumpkin rootstock grafted plants were also found to have fruit with similar aroma characteristics to self-grafted plants (Verzera et al., 2014). It was reported that grafting with pumpkin rootstock significantly reduced the activities of ADH and AAT in oriental melon fruit, and reduced the aroma of melon fruit (Hao et al., 2018). In this study, the odor score of RG fruits was significantly lower than that of SG fruits (Table 2). Notably, butanoic acid ethyl ester, 2-methyl-2-butene-1-al, and 2-methylheptan-1-al, which are among the main volatiles in SG fruits, could not be detected in RG fruits (Table 4). However, 32 volatile compounds were identified in RG fruit that were not present in SG fruit in which (E,E)-2,4-heptadienal is described as having a nutty flavor, octanoic acid has a sweaty smell, styrene has gasoline smell, citral and trans, cis-2,6-nonadiene-1-ol are noted for their lemon and cucumber flavors, respectively (Table 4). In addition, there were significant differences in the relative contents of 24 volatile compounds in RG and SG fruits. Among the compounds with increased relative content in RG fruits, 2-methyl-1-butanoacetate and 2-pentylfuran are described as having a fruity flavor, whereas 1-nonanol and 1-heptanol are characterized by grass flavor and cucumber flavor (Table 6). However, only the odors of eucalyptol and farnesene of the 10 compounds with significantly lower contents in RG fruits have previously been described, both of which are said to impart a sweet aroma. The type and content difference of volatile compounds caused the reduced fruit quality after grafting using pumpkin rootstocks.

At 30 DAP, the ADH activity of RG fruits was significantly lower than that of SG fruits, but the AAT activity has no difference compared with SG fruits. After fruit ripening (42 DAP), the ADH activity decreased, and the AAT activity reached 4.8−8.1 times of 30 DAP. The ADH activity of RG fruits has no difference with that of SG fruits, but the AAT activity was significantly lower than that of SG fruits (Fig. 4). Chen showed that strong-aromatic melon cultivar ‘Cai Hong’ had significantly higher AAT activity than the non-aromatic melon ’Cai Gua’ (Chen et al., 2016). Therefore, the reduced AAT activity of RG fruits may be the physiological reason for the reduced aroma of ‘Qingxian Yangjiaocui’ melon fruits. Manri’quez found that CmADH1 and CmADH2 are involved in the synthesis of volatiles in fruits, and CmADH1, CmADH2 proteins may play a specific role in providing substrates for downstream AAT (Manríquez et al., 2006). Previous study reported that CmAAT played an active role in the synthesis of esters in melon fruits (Galaz et al., 2013). The expression levels of CmADH1 and CmADH2 were significantly lower in RG fruits than in SG fruits at 30 DAP and 42 DAP, while the expression levels of CmAAT1 and CmAAT2 were significantly lower in SG fruits at 42 DAP (Fig. 5). Therefore, the reduced expression levels of CmAATs and CmADHs homologs may be another important reason for the reduced aroma quality of fruits from grafted plants on pumpkin rootstocks.

The broad implications of rootstock selection in horticulture

Selection of suitable rootstocks is an important factor in the improvement of horticultural crop traits based on grafting technology. At present, it has been confirmed that different rootstock types significantly affect the aroma composition of the fruit in peach, citrus, tomato and melon (Benjamin, Tietel & Porat, 2013; Jukić Špika et al., 2021; Seker, Ekinci & Engin, 2017; Verzera et al., 2014). In this study, grafting with muskmelon rootstocks had little effect on fruit aroma, whereas grafting of pumpkin rootstocks reduced or even eliminated the content of the main aromatic volatiles in melon fruit, with the appearance of bad odor volatiles, and significantly reduced the activity of ADH and AAT and the expression level of related genes in the fruit, reducing fruit aroma quality. These results suggested that grafting with improper rootstocks led to the reduced expression level of CmADH and CmAAT homologs, and then ADH and AAT activities were significantly decreased, which respectively restrict or even blocked the synthesis of alcohols and ultimate esters. Esters were the main volatile compounds contributing to the aroma of ‘Qingxian Yangjiaocui’ melon fruits so that the flavor quality of melon fruits was reduced and resulted in a dissatisfied consumer perception after grafting with unsuitable rootstocks such as some pumpkin rootstocks. On the contrary, muskmelon rootstocks had little effect on the expression level of CmADH and CmAAT homologs, and subsequent ADH and AAT activities so that the flavor quality of QG fruits was similar to SG fruits. Therefore, the closer the relationship between rootstock and scion is, the more beneficial it is to improve the fruit aroma quality which may due to the enhanced grafting compatibility.

The results of this study revealed the physiological and molecular mechanisms underlying the effects of different genotypes of rootstocks on melon fruit aroma, and laid a theoretical foundation for the selection of rootstocks and the regulation of melon fruit aroma quality in future melon production. However, the varieties of rootstocks were limited in this study, and it is failed to further study the interaction between rootstock and scion in this study. The differences in vascular connectivity and signaling between different rootstock and scion combinations have a significant effect on the transport of water, nutrients, and phytohormones (Fallik & Ziv, 2020). Future studies about mobile proteins or small RNAs within the phloem will contribute to reveal the interaction mechanism between rootstock and scion, and explore the influence mechanism of flavor quality by grafting in melon.

In SG, QG and RG fruits, 211, 211 and 216 volatiles were respectively identified, and these compounds included esters, alcohols, hydrocarbons, aldehydes, ketones, nitrogen compounds, phenolics, acids, and sulfur compounds (Table S2). However, the regulatory networks among detected volatiles and the main factors involving in the regulation of melon fruit aroma quality need to be further studied. Although ADH and AAT activities were detected in this study, the activity of key enzymes regulating the synthesis of other aromatic substances except for esters were still unclear. In addition, the specific functions of key genes such as CmADH and CmAAT homologs were unknown in melon, though the expression levels of CmADH and CmAAT were changed by the application of different rootstocks. Future studies using overexpression or CRISPR/CAS system to obtain stable transgenic lines of genes encoding key aromatic volatiles, would be promising to effectively improve the flavor quality of melon.

Conclusion

Esters, alcohols, and aldehydes were the main aroma components of ‘Qingxian Yangjiaocui’ melon fruits. Grafting with pumpkin rootstocks significantly reduced the odor preference scores of melon fruits, decreased the content or even caused the absence of main aroma components, and produced the volatile compounds with unpleasant odor. In addition, grafting with pumpkin rootstocks significantly decreased ADH and AAT activity and the expression levels of CmADH and CmAAT homologs in SG fruits. However, grafting with muskmelon rootstocks had no significant effect on fruit aroma, ADH and AAT activities, and the expression of CmADH and CmAAT homologs. The detection of four new aromatic volatiles with a sweet smell in QG fruits compared to SG fruits highlights the potential benefits of grafting with muskmelon rootstocks in enhancing aroma. This study revealed the physiological and molecular mechanisms underlying the effects of different genotypes of rootstocks on melon fruit aroma by combining volatile compounds identification, enzyme activities detection and expression analyses. In future, muskmelon rootstocks with strong growth potential and excellent disease resistance were recommended to applied to effectively reduce the negative impact of grafting on flavor quality in melon production.

Supplemental Information

Supplemental Information 1 MIQE checklist

Click here for additional data file.

Supplemental Information 2 Statistical Analysis by R

Click here for additional data file.

Supplemental Information 3 Raw data of GC-MS for SG

Click here for additional data file.

Supplemental Information 4 Raw data of GC-MS for QG

Click here for additional data file.

Supplemental Information 5 Raw data of GC-MS for RG

Click here for additional data file.

Supplemental Information 6 Peak values of volatile substances in the fruits of three types of rootstock grafted plants

SG: self-grafted; QG: grafted on ‘Qinmi No. 1’; RG: grafted on ‘Ribenxuesong’.

By comparing the peak time and peak area of volatile compounds, the properties and content of volatile compounds in fruits and melons can be determined. The activities of alcohol dehydrogenase (ADH) and alcohol acyltransferase (AAT), as well as the expression levels of related genes, are shown in Table S4 and Table S5, respectively.

Click here for additional data file.

Supplemental Information 7 Volatile compounds detected in different melon fruits

SG, self-grafted; QG, grafted on ‘Qinmi No. 1’; RG, grafted on ‘Ribenxuesong’.

Click here for additional data file.

Supplemental Information 8 Eigenvectors of PCA

Click here for additional data file.

Supplemental Information 9 Activity of alcohol dehydrogenase (ADH) and alcohol acyltransferase (AAT) in melon fruit

SG, self-grafted; QG, grafted on ‘Qinmi No. 1’; RG, grafted on ‘Ribenxuesong’.

Click here for additional data file.

Supplemental Information 10 The expression levels of alcohol dehydrogenase (CmADH1, CmADH2) and alcohol acyltransferase (CmAAT1, CmAAT2) related genes in melon fruits

SG, self-grafted; QG, grafted on ‘Qinmi No. 1’; RG, grafted on ‘Ribenxuesong’.

Click here for additional data file.

Additional Information and Declarations

Competing Interests

Author Contributions

Data Availability

The authors declare there are no competing interests.

Kedong Guo conceived and designed the experiments, performed the experiments, analyzed the data, prepared figures and/or tables, authored or reviewed drafts of the article, and approved the final draft.

Jiateng Zhao performed the experiments, prepared figures and/or tables, and approved the final draft.

Siyu Fang performed the experiments, prepared figures and/or tables, and approved the final draft.

Qian Zhang performed the experiments, prepared figures and/or tables, and approved the final draft.

Lanchun Nie conceived and designed the experiments, performed the experiments, analyzed the data, authored or reviewed drafts of the article, and approved the final draft.

Wensheng Zhao conceived and designed the experiments, performed the experiments, analyzed the data, authored or reviewed drafts of the article, and approved the final draft.

The following information was supplied regarding data availability:

The raw data of GC-MS is available at EMBL-EBI MetaboLights: MTBLS8677.

https://www.ebi.ac.uk/metabolights/MTBLS8677.

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
