# Peer review of "The effects of different rootstocks on aroma components, activities and genes expression of aroma-related enzymes in oriental melon fruit"

_PeerJ, doi:10.7717/peerj.16704_

## Round 0.1 · original submission · Major Revisions

The authors are requested to revise the manuscript as per suggestions of the reviewers.

**Language Note:** The review process has identified that the English language must be improved. PeerJ can provide language editing services - please contact us at [email protected] for pricing (be sure to provide your manuscript number and title). Alternatively, you should make your own arrangements to improve the language quality and provide details in your response letter. – PeerJ Staff

Reviewer 1 ·

Basic reporting

The study explores the impact of grafting melon plants onto different rootstocks (Cucumis melo L. and Cucurbita maxima Duch.) on fruit aroma quality. The research investigates changes in odor intensity, preference scores, and the composition of volatile compounds. While the study provides valuable insights into the influence of rootstocks on melon fruit aroma, there are several aspects that require clarification, expansion, or refinement.
Comments
• The evaluation of odor intensity and odor preference scores provides a quantitative measure of aroma quality.
• In the abstract section the author should add the future implication of this study.
• The detection of four new aromatic volatiles with a sweet smell in self-grafted plants compared to non-self-grafted plants highlights the potential benefits of grafting in enhancing aroma.
• The paper should be organized more logically and comprehensive manner. It is essential to clearly outline the research objectives, methods, results, and conclusions in a structured manner.
• In the introduction section please provide a comprehensive background on grafting in melon cultivation and its significance for fruit quality, setting the context for the specific research questions addressed.
• Ln 98-101: Please rewrite the content.
• There should be a clear statement of the objectives and hypotheses of the study. What specific hypotheses were tested in this study and why? This has to be highlighted in the introductions section.
• The introduction section is too short. Kindly improve it.
• The section “Evaluation of the odors of mature fruits from self-grafted plants and plants with different rootstock” needs to be revisited and write the result should be in more detailed form.
• Clarify how odor intensity and preference scores were assessed. Were these sensory evaluations conducted by a panel of trained assessors or consumers?
• The analysis of volatile compounds should include information on the analytical techniques used for detection and quantification, such as gas chromatography-mass spectrometry (GC-MS).
• It's crucial to discuss the biological relevance of the observed changes in aroma compounds. How might these alterations affect the overall aroma profile and consumer perception of melon fruits?
• In discussion section, Consider discussing potential limitations and sources of variation in the study, which could affect the interpretation of the results. Address the broader implications of rootstock selection in horticulture and whether similar effects might be observed in other fruit crops.
• The conclusion should provide a concise summary of the key findings and their practical implications of the research work performed.
• Carefully proofread the manuscript for language and grammar issues to enhance clarity and readability.

Experimental design

Please see the comments in section 1

Validity of the findings

Please see the comments in section 1

Additional comments

Please see the comments in section 1

Reviewer 2 ·

Basic reporting

Language and Clarity: While the manuscript provides valuable insights, some sentences could benefit from rephrasing for clarity. For instance, “…were disappeared, while…” (Line 71) might be revised as “…were absent…”. "Melons can be classified as either aromatic or non-aromatic types..." (Line337) might be clearer as "Melons are typically classified into aromatic and non-aromatic types based on their fruit aroma intensities." Additional language editing and proofreading of the manuscript are strongly recommended.
Introduction & Background: The context provided is good, but the introduction would benefit from pinpointing the knowledge gaps this study specifically addresses. This will help readers quickly discern the study's unique contribution.
Literature Citations: The current references are relevant, but their integration feels somewhat repetitive. Consider focusing on the most critical studies that directly influenced this work, ensuring each citation adds unique value.
Structure & Organization: The manuscript would benefit from clearer subheadings, especially in the discussion section. This would help readers navigate the content more easily.
Figures & Tables: The figures and tables appear relevant. However, better labeling, especially for Figure 2 would enhance clarity. The authors may consider using zoomed view for TICs and label the time axis in using integers.

Experimental design

Originality: The research is primary and seems to be a continuation or a deeper exploration of existing research, particularly focusing on the 'QingxianYangjiaocui' melon variety.
Research Question: The manuscript hints at its research questions but making them explicit would help readers understand the study's goals. A clear statement in the introduction would be beneficial. the manuscript seems to address the following research questions:
• How can melons, specifically 'QingxianYangjiaocui', be classified based on their aromatic components?
• What are the main volatile compounds contributing to the aroma of 'QingxianYangjiaocui' melon fruits?
• How does grafting, specifically with muskmelon and pumpkin rootstocks, influence the aroma components and overall fruit quality of 'QingxianYangjiaocui' melon?
• What are the physiological and molecular mechanisms underlying the effects of different genotypes of rootstocks on melon fruit aroma?
Rigorous Investigation: The research appears to have been conducted rigorously. However, the manuscript does not provide explicit details on the control conditions, which would be essential for understanding the complete experimental setup.
Methodology: The methods, particularly the HS-SPME-GC-MS technique, are described, but there are concerns about the quantitation approach. The "relative contents" derived from GC-MS profiles can be ambiguous without the use of reference standards. It's crucial to quantify metabolites using reference standards or validate identified metabolites' GC-MS profiles using commercial synthetic counterparts for accurate results.

Validity of the findings

Data Quality: The underlying data seems to be robust, but the statistical soundness might be compromised if the GC-MS data hasn't been validated with reference standards. The use of relative content without clear reference compounds can lead to misinterpretation.
Comparison with Existing Literature: The discussion contrasts the findings with existing knowledge, but delving deeper into how this study advances or challenges previous findings would strengthen its impact.
The conclusion section is mostly aligned with the findings, but being careful to avoid overreaching statements is essential. For instance, when discussing the impact of grafting on aroma, ensure that all claims are backed by the presented data.

Additional comments

This manuscript delves deep into the nuances of melon aroma components, particularly highlighting the 'QingxianYangjiaocui' variety. The study's strengths are evident in its meticulous approach and concentrated subject matter. However, there's room for improvement in language clarity, more seamless integration of related literature, and enhancing the data's robustness. The insights presented on the aromatic classifications of melons are invaluable, especially regarding the influence of grafting. While the research framework is commendably structured and buttressed by extensive literature references, the GC-MS quantitation methodology raises concerns. Addressing these concerns is imperative to ensure the validity of the findings and for the manuscript to be deemed fit for publication.

Reviewer 3 ·

Basic reporting

In this manuscript entitled “The effects of different rootstocks on aroma components, activities and genes expression of aroma-related enzymes in oriental melon fruit” by Guo et al., the authors explored the choices of rootstocks in the grafting of melons. Overall, this is a comprehensive study that evaluates the effect of using different rootstocks. The authors reported order intensity, odor preferences, GC-MS analysis, enzyme activity, and gene expression. The topic has high economic values in the cultivation of melons, and would be of particular interest for the melon industry.
Major suggestions:
1. In the introduction section, it would be beneficial to thoroughly introduce the current knowledge about the aroma compounds in melon, and in other similar fruits.
2. In addition, it would be necessary to thoroughly introduce the synthesis pathway that ADH and AAT are involved in.
3. For the statistical analysis in R, please include the R scripts (.r) or R markdown (.rmd) files in the supplementary materials, to ensure reproducibility of the analysis, and to comply with the FAIR standard of the journal policy.
4. Please include phenotypic pictures of the melon fruits grafted to different rootstocks in this study. This will help the readership to know whether fruit shapes and color are changed.
5. It is “principal component analysis”, not “principle”. Please fix this in every occurrence. https://en.wikipedia.org/wiki/Principal_component_analysis
6. For the principal component analysis shown in Figure 3, please include a table showing the eigenvectors. Without knowing the eigenvectors, the PCA plot is just some datapoints, not effectively exhibiting the scientific discoveries. To help the authors determine which table should be added, please refer to the first table in this link: https://support.minitab.com/en-us/minitab/21/help-and-how-to/statistical-modeling/multivariate/how-to/principal-components/interpret-the-results/key-results/
7. All figures are in poor resolution. Please regenerate all figures with higher resolution.
8. According to the journal policy, all the raw data need to be properly deposited into public-accessible databases. The GC-MS raw spectrums need to be deposited into an appropriate database with access code provided in the manuscript, for example, GOLM METABOLOME DATABASE (http://gmd.mpimp-golm.mpg.de/).
Minor suggestions:
1. Line 92-93, “Alcohol dehydrogenase (ADH) and alcohol acyltransferase (AAT) and their related genes are involved in the last two steps of volatiles synthesis in fruit.” Please write which specific volatile compounds are referred to in this sentence.
2. It would be helpful to include the native language of the melon cultivars in the methods section, or at the first occurrences, if allowed by the journal. As this research have application values in the melon industry, including cultivar name in the native language in addition to pinyin would help the growers.
3. The sentences in the methods should follow similar structures. For example, line 191-193, “Grind 3 g of frozen fruit sample in liquid nitrogen, add 2.25 mL of protein extract and extract on ice for 20 min, then centrifuge at 12,000 g for 20 min at 4 °C” is not consistent with the structure of other sentences. This sentence should be rewritten. Please check other sentences in the methods section and fix similar issues.
4. Line 204, “))” should be “)”. Please check carefully for copy-editing issues.

Experimental design

See above.

Validity of the findings

See above.

---

## Round 0.2 · Minor Revisions

The authors are requested to revise the manuscript as per the suggestions of reviewer 3.

Reviewer 1 ·

Basic reporting

The authors made significant changes in the manuscript as per my suggestion.

Experimental design

See section 1

Validity of the findings

See section 1

Additional comments

See section 1

Reviewer 2 ·

Basic reporting

No more comments

Experimental design

No more comments

Validity of the findings

No more comments

Additional comments

The authors have comprehensively addressed all the issues I previously raised. I endorse the publication of this manuscript.

Reviewer 3 ·

Basic reporting

In the revised manuscript entitled “The effects of different rootstocks on aroma components, activities and genes expression of aroma-related enzymes in oriental melon fruit”, the authors addressed most of the reviewers’ concerns. However, there are a few points not being addressed.
1. The reviewer requested the authors to deposit the GC-MS raw spectrums into public databases. The authors stated that “The raw data of GC-MS have been deposited to a public metabolomics repository MetaboLights (https://www.ebi.ac.uk/metabolights) under the accession number of MTBLS8677”. However, upon searching with the accession number MTBLS8677, the dataset cannot be found. Please make sure to set the data as publicly accessible before publication.
2. The reviewer requested the authors to correct “Principle component analysis” to “Principal component analysis”. The title of Figure 3 was not corrected.

Experimental design

No comment.

Validity of the findings

No comment.

---

## Round 0.3 · accepted · Accept

The manuscript can be accepted in its current state.